# Experimental Investigation on the Machinability of PCBN Chamfered Tool in Dry Turning of Gray Cast Iron

Ganggang Yin [1,2,*], Jianyun Shen [3], Ze Wu [2], Xian Wu [3,*] and Feng Jiang [4]

1   Kistler Innovative Technology China Ltd., Shanghai 201107, China
2   School of Mechanical Engineering, Southeast University, Nanjing 211189, China
3   College of Mechanical Engineering and Automation, Huaqiao University, Xiamen 361021, China
4   Institute of Manufacturing Engineering, Huaqiao University, Xiamen 361021, China
*   Correspondence: ganggang.yin@kistler.com (G.Y.); xianwu@hqu.edu.cn (X.W.)

**Abstract:** Polycrystalline cubic boron nitride (PCBN) tools are widely used for hard machining of various ferrous materials. The edge structure of the PCBN cutting tool greatly affects the machining performance. In this paper, dry turning experiments were conducted on gray cast iron with a PCBN chamfered tool. Both the cutting temperature and the cutting force were measured, and then the surface quality and tool wear mechanisms were analyzed in detail. It was found that the cutting temperature and cutting force increased with the increase in feed rate, depth of cut, and cutting speed. The surface roughness firstly decreased, and then increased with an increase in feed rate. The minimum surface roughness was obtained with a feed rate of 0.15 mm/r which exceeded the tool chamfer width. The PCBN tool wear mode was mainly micro notches on the rake face and micro chipping on the tool chamfer, while the adhesion wear mechanism was the main tool wear mechanism.

**Keywords:** PCBN tool; gray cast iron; surface quality; tool wear

## 1. Introduction

Cast iron materials are widely used in the automotive industry, e.g., in engine boxes and brake discs [1,2]. In terms of hard machining cast iron, Martinho et al. investigated the continuous dry turning of gray cast iron using ceramic tools with and without diamond coating; the coated tools presented higher main cutting force and lower flank wear [3,4]. Heck et al. found the different wear behaviors of PCBN cutting tools during machining of compacted graphite iron compared to gray cast iron [5]. Grzesik et al. complexly assessed cutting force, cutting temperature, tool wear progress, and surface roughness in orthogonal turning of nodular cast iron with CBN tools with a chamfer width of 100 μm [6]. Chen et al. studied the influence of material properties on the machining performance in machining of high-chromium white cast iron with CBN tools, and it was found that the micro hardness of workpiece material showed significant impacts on the tool wear, tool life, cutting forces and surface quality [7].

The PCBN material has many excellent properties, such as high hardness, high temperature resistance, good chemical stability, and low friction coefficient [8,9]. These excellent properties make it become the most perfect cutting tool material for hard machining of various steel materials [10,11], such as hardened steel, cast iron, bearing steel, and superalloys. As one of the super-hard cutting tools, PCBN cutting tools have been widely applied in aerospace, automobile, and construction machinery manufacturing. There is extensive literature focused on the fabrication and applications of PCBN cutting tools [12,13]. Regarding the preparation of CBN tool materials, Yun et al. designed and fabricated CBN tool materials with a graded structure to improve material strength and toughness [14]. Mo et al. found that the increase in CBN percentage increased the hardness and wear resistance of PCBN composite, and then increased tool life in the cutting of hard-bearing steel GCr15 [15].

Tamang et al. investigated the brazing of CBN material to tungsten carbide by microwave hybrid heating by applying a paste of Ag–Cu–In–Ti alloy [16]. Fei et al. attempted the pulsed magnetic treatment on CBN tools to improve the cutting performance and found a relatively lower cutting force and less wear area [17]. In terms of the applications of CBN cutting tools, Manoj et al. experimentally studied the machinability of AISI D6 tool steel using low-percentage-CBN tools. They analyzed the effect of machining parameters on cutting temperature, cutting force, and surface roughness, and they found that crater wear, micro chipping, and cutting edge breakage were the main tool wear mechanisms during continuous turning [18]. Gutnichenko et al. investigated the hard turning of helical gear hubs made of low-carbon alloyed steel after heat treatment with hardened up to HRC 60–63 using PCBN cutting tools; it was found that the main tool wear characterizations were flank wear and crater formation, and different strategies were also proposed to monitor tool wear and predict tool life [19]. Schultheiss et al. evaluated the tool wear mechanisms in the machining of gray cast iron with PCBN tools, the results showed that flank and notch wear were the main wear mechanisms, and the non-chamfered tools exhibited the higher notch wear [20].

The tool cutting edge geometry greatly affects the cutting performance, such as the tool nose radius and cutting edge passivation [21,22]. Kumar et al. investigated the effect of cutting parameters and tool geometry in hard machining of hardened AISI H13 material; it was found that the surface roughness decreased while the cutting forces increased with increasing tool nose radius [23]. The chamfered tool structure has a narrow edge ground on the tool rake face with a negative rake angle to enhance tool cutting edge strength. A chamfered tool can improve tool life and is widely used in the cutting tool treatment for rough machining [24,25]. Grzesik et al. developed an empirical model of friction for oblique cutting with CBN chamfered tools; it was found that the chamfered CBN tool decreased the friction coefficient [26]. Tang et al. developed a special chamfer on cutting tools and used it to suppress tool wear in the machining of hard material [27]. Souza et al. investigated the effect of edge preparation of PCBN tools on the wear performance during orthogonal turning of high-speed steels; the PCBN tools with ground land and honed edge radius, as well as a high CBN content, showed chipping in the rake face [28]. Ventura et al. studied the effect of cutting edge geometry on the tool wear behavior of CBN tools during interrupted turning of hardened steel; the results indicated that the chamfered cutting edge reinforces the cutting edge without excessively increasing the mechanical and thermal loads [29]. In summary, a reasonable tool geometry design is helpful to obtain better machining quality and tool life.

In this paper, to assess the machinability of the PCBN chamfered tool in hard machining of gray cast iron, dry turning experiments were performed on gray cast iron with the PCBN cutting tools. The effect of machining parameters on the cutting temperature and cutting force was measured. Then, the machined surface roughness and tool wear were studied in detail. The related machining mechanisms were analyzed to understand the cutting process according to the experimental results. The results are helpful for machining parameter selection in the machining of brittle materials with PCBN cutting tools.

## 2. Experimental Procedure

Due to their excellent properties, PCBN tools are commonly used in the machining of cast iron materials. The used cutting tool for rough machining in this work was the PCBN cutting tool (grade BN500, Sumitomo, Tokyo, Japan) prepared by Xiamen Xiazhi Technology Tool Co., Ltd., Xiamen, China, as shown in Figure 1. The grain size of CBN materials was about 1–2 μm, and the measured volume proportion of CBN material in the PCBN tools was about 56%. The PCBN tool presented a nose radius of 1.5 mm and cutting edge radius of 25 μm. The tool rake angle and clearance angles were 0° and 5°, while the cutting edge angle is 45°. The tool chamfer with a width of 0.1 mm and a negative rake angle of −20° was prepared on the rake face of the PCBN tool, as shown in Figure 1a.

The tool chamfer is helpful to enhance the cutting edge, and then avoid the premature chipping of the tool cutting edge in the machining of the brittle materials.

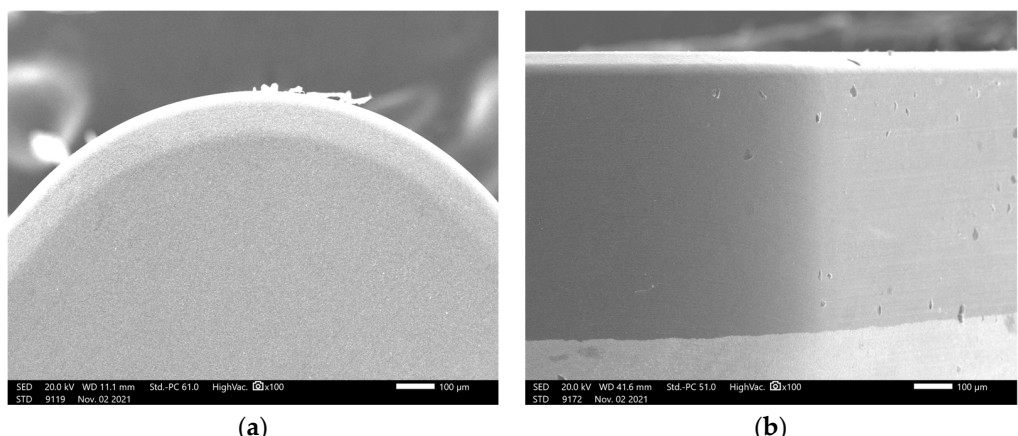

**Figure 1.** PCBN cutting tool: (**a**) rake face; (**b**) flank face.

Dry turning experiments were conducted on a CNC turning machine tool (CAK6150, Shenyang Machine Tool Co., Ltd., Shenyang, China) as shown in Figure 2a. This machine tool is widely used to manufacture rotary parts in various applications. The maximum spindle rotating speed was 2000 rpm. The workpiece material was gray cast iron HT200, which is widely used for the brake disc and engine cylinder in the automotive industry [20,30]. The material properties of the workpiece are listed in Table 1. The workpiece samples were cut into small pieces with dimensions of Ø 50 × 250 mm, and then clamped in a three-jaw chuck machine tool. In the turning experiments, the cutting temperature was measured using a thermal imager (TIM 200, Micro Epsilon, Germany), with a temperature measurement range of −20–900 °C. The cutting force was recorded using a dynamometer (9257B, Kistler, Winterthur, Switzerland), which is widely used in various applications for cutting force measurement [31]. The cutting force measurement system was composed of a dynamometer, connecting cable, charge amplifier (5080A, Kistler, Winterthur, Switzerland), DAQ system, and PC, as shown in Figure 2b. The sampling frequency used in cutting force measurement was 20 kHz to obtain the complete cutting force signal without distortion.

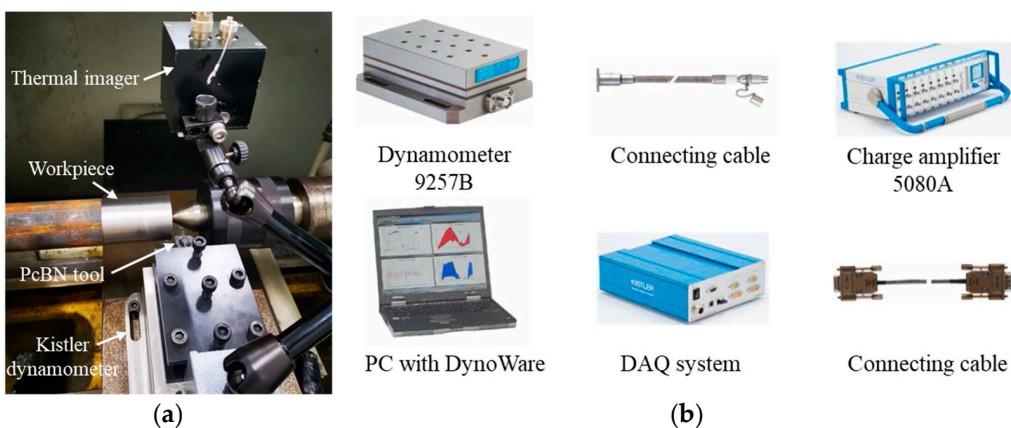

**Figure 2.** Experimental setup: (**a**) turning experiment, (**b**) cutting force measurement system.

In the experiment, the workpiece was pre-machined to ensure a smooth surface using a cemented carbide tool [32], and then the continuous turning experiments were performed. The detailed turning parameters are shown in Table 2. In single-factor experiments, four different levels were set for the machining parameters of spindle speed, feed rate, and depth of cut, according to the existing literatures [10,33]. The corresponding cutting speeds for different spindle rotating speeds were 110, 157, 204, and 251 m/min. The experiments

were divided into two stages; during single-factor turning experiments in the first stage, the feed travel for every level of machining parameters was 20 mm, while the total feed travel for every variable of the machining parameters was 80 mm. Then, to test the tool wear mechanism, the PCBN cutting tool was used for the continuous tool wear experiment to reach the feed travel of 4000 mm with fixed machining parameters ($v$ = 204 m/min, $a_p$ = 0.2 mm, and $f$ = 0.15 mm). After the experiment, the workpiece was cleaned using an ultrasonic cleaner, and then the surface roughness was inspected with a roughness meter (3220, Beijing Shidaizhifeng Instrument Co., Ltd., Beijing, China) at different positions along the feed direction. The sampling length and cutoff wavelength were 2.5 mm in the surface roughness measurement. Each experiment was repeated three times, and the averaging values were adapted as the results. The tool wear was inspected using a scanning electron microscope (SEM, JSM-IT500, JEOL, Tokyo, Japan).

**Table 1.** Material properties of gray cast iron HT200.

| Properties | Value |
|---|---|
| Density (g/mm$^3$) | 7.2 |
| Hardness (HB) | 193 |
| Tensile strength (MPa) | 320 |
| Elasticity modulus (GPa) | 135 |
| Poisson ratio | 0.3 |

**Table 2.** Turning parameters.

| Parameters | Value |
|---|---|
| Spindle speed $n$ (rpm) | 700, 1000, 1300, 1600 |
| Depth of cut $a_p$ (mm) | 0.1, 0.2, 0.3, 0.4 |
| Feed rate $f$ (mm/r) | 0.05, 0.1, 0.15, 0.2 |

## 3. Results and Discussions

### 3.1. The Cutting Temperature

In addition to the thermal image, the used thermal imager provides a camera to record a video image of the cutting process, as shown in Figure 3. The emissivity of the workpiece material was calibrated to about 0.5 in this work using a digital display heating platform. Combining the thermal image and video image, it was easier to observe the temperature field distribution of the cutting process. It was observed that the chips formed at the cutting zone with fragment shapes and were then sprayed around. In the recorded thermal image, the maximum temperature is exhibited in the cutting zone, and the sprayed chips present a second temperature source with heat dissipation during the spraying process. The mean temperature in the contact zone between the tool and workpiece is difficult to acquire; hence, during the turning experiment, the maximum temperature measured in the cutting zone and displayed in the upper right corner was adopted as the cutting temperature.

The cutting temperature dramatically affects the cutting process and the machining quality. Figure 4 shows the effect of machining parameters on the measured cutting temperature. The results show that the cutting temperature increased from 226 °C to 429 °C with the feed rate increasing from 0.05 mm/r to 0.2 mm/r. Its increasing rate was rapid at first but decreased for a feed rate larger than 0.1 mm/r. The cutting temperature increased from 278 °C to 423 °C with the cutting speed increasing from 100 m/min to 251 m/min. The initial increasing rate slowed after the cutting speed exceeded 157 m/min. This was probably due to the the heat dissipation area being similar under the same depth of cut when the cutting temperature was relatively higher with more extensive machining parameters, while the temperature difference with the surrounding environment became larger. Hence, the larger temperature difference caused more heat dissipation and resulted in a relatively low temperature increase. The cutting temperature almost linearly increased

from 257 °C to 369 °C with an increase in the depth of cut, while the heat dissipation area also increasing with the depth of cut.

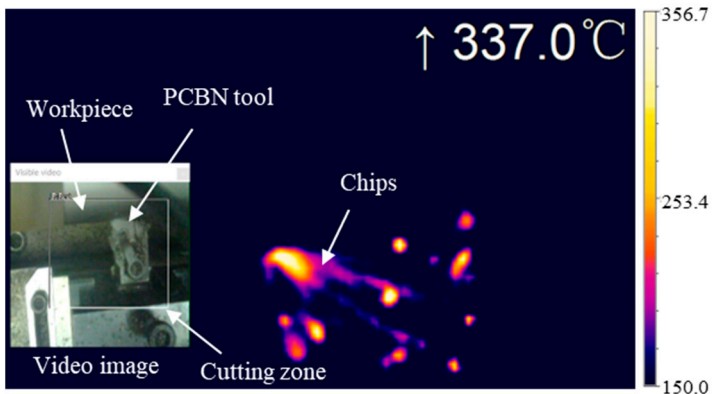

**Figure 3.** Cutting temperature measurement.

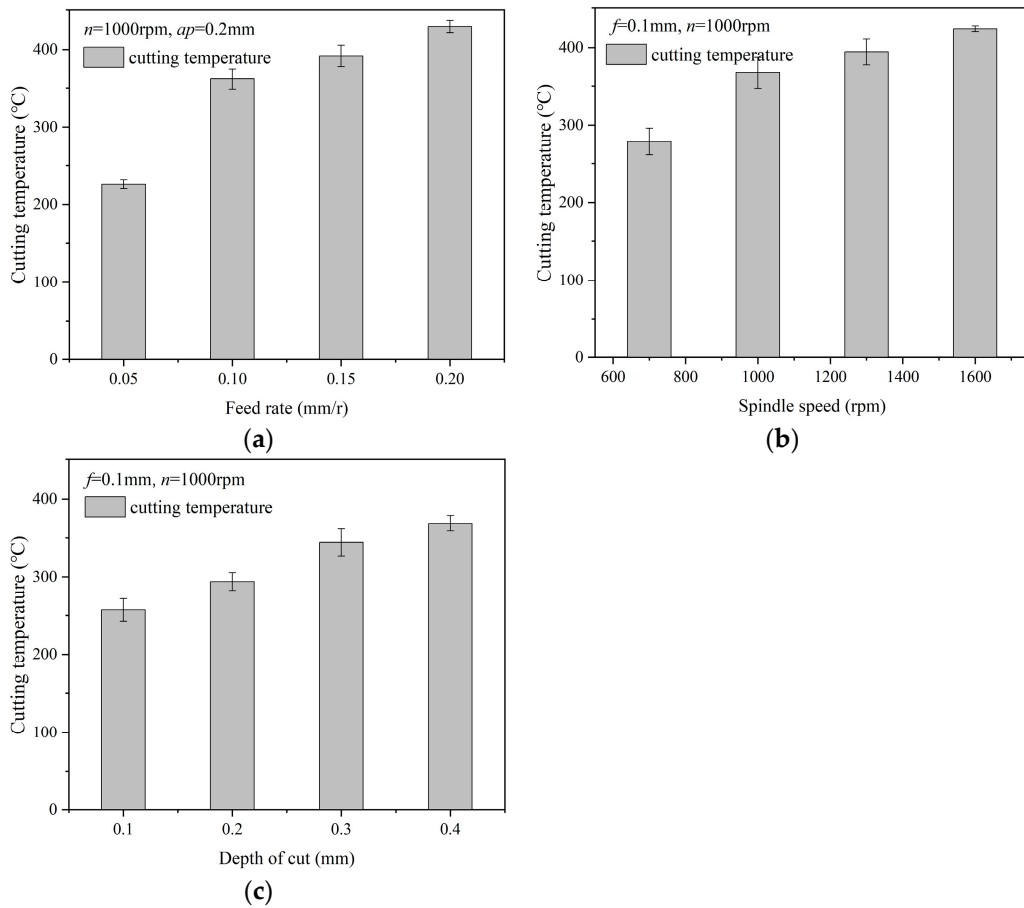

**Figure 4.** Effect of machining parameters on cutting temperature: (**a**) effect of feed rate on cutting temperature; (**b**) effect of cutting speed on cutting temperature; (**c**) effect of depth of cut on cutting temperature.

### 3.2. The Cutting Force

Figure 5 shows the recorded cutting force signal in continuous turning experiments with different feed rates. The three cutting force components in order of magnitude were the tangential force (main cutting force) $F_z$, the radial force (passive force) $F_y$, and the axial force (feed force) $F_x$. The radial force $F_y$ was similar to the tangential force $F_x$. In the turning process, the depth of cut was much lower than the tool nose radius, and only

a small segment of the tool nose arc took part in cutting process. The contact condition resulted in the equivalent tool cutting edge angle becoming small. Hence, the component of the resultant cutting force in the radial direction Fy increased. From the results, the cut in and cut out process can be clearly observed, and the cutting force signal waveform was more stable under a large feed rate than a low feed rate of 0.05 mm/r. Under the low feed rate of 0.05 mm/r, the profound ploughing effect led to an unstable cutting process because of the small, undeformed chip thickness. The unstable cutting process resulted in noticeable fluctuations in the cutting force signal. The mean value of the cutting force in the stable cutting process was adopted to quantitatively analyze the cutting force results.

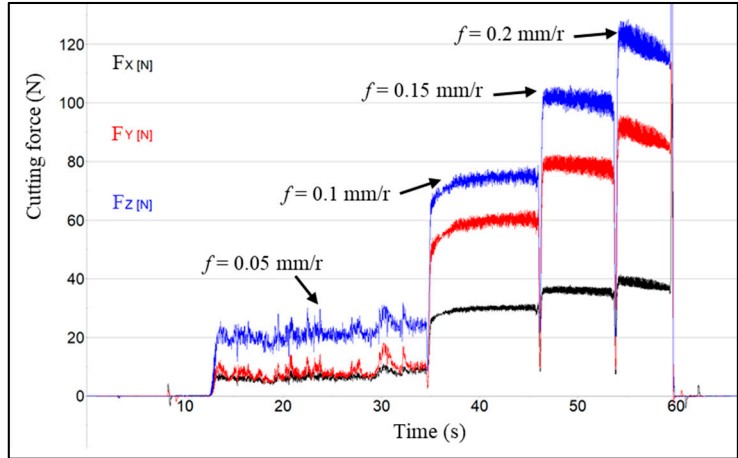

**Figure 5.** The cutting force signal.

Figure 6 shows the cutting force varying with the machining parameters. The cutting forces Fx, Fy, and Fz increased from 6 N, 8 N, and 20.5 N to 36.5 N, 86.5 N, and 115.5 N, respectively, with the increase in feed rate. The increment in axial force Fx was much smaller than that in tangential force Fz and radial force Fy. A similar phenomenon to the cutting temperature was observed, whereby the increasing rate was rapid at first but decreased for feed rates larger than 0.1 mm/r. The cutting process was continuous turning in this work, and the adhesion material on the tool surface was small at the first stage. A lower tool adhesion caused a relatively low cutting force.

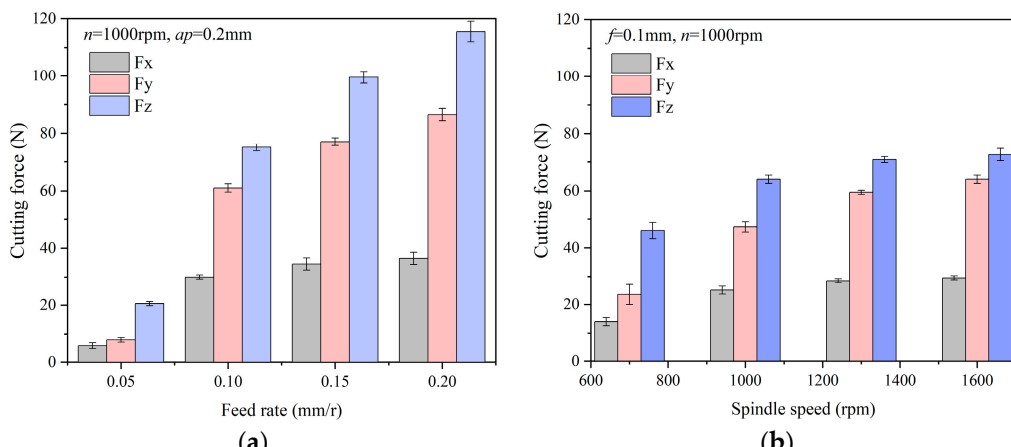

**Figure 6.** *Cont.*

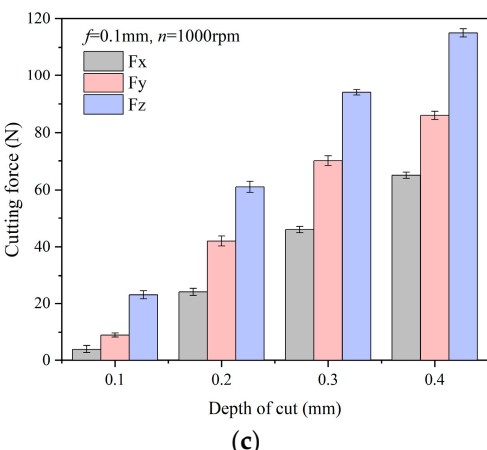

(**c**)

**Figure 6.** Effect of machining parameters on cutting force: (**a**) effect of feed rate on cutting force; (**b**) effect of cutting speed on cutting force; (**c**) effect of depth of cut on cutting force.

Although the cutting temperature increased with an increase in cutting speed, the cutting force increased gradually. A probable reason is that this cutting temperature was in the built-up edge (BUE) formation scope. A serious BUE in the cutting process increases the friction and material removal, resulting in a larger cutting force. The cutting speed should be further increased to avoid the BUE. The cutting forces Fx, Fy, and Fz increased from 4 N, 9 N, and 23 N to 65 N, 86 N, and 115 N, respectively, when the depth of cut increased from 0.1 mm to 0.4 mm. This is similar to the cutting temperature, whereby the cutting force linearly increased with an increase in the depth of cut.

### 3.3. The Surface Quality

Figure 7 shows the surface roughness Ra and surface morphology under different feed rates. It was found that the surface roughness firstly decreased and then increased with an increase in feed rate. The built-up edge was formed on the tool rake face due to the adhesion materials. When the built-up edge along the tool nose arc was uneven, the actual undeformed chip thickness along the tool nose arc was also uneven. Then, the uneven undeformed chip thickness led to transverse micro tears perpendicular to the cutting direction [34]. These micro tears on the surface morphology confirmed the formation of built-up edge in the cutting process.

Surface roughness is determined by the physical factors and theoretical residual area, which is proportional to the feed rate and tool geometry. As shown in Figure 8, the main physical factors significantly influencing the surface roughness included micro burrs and micro bulges induced by the ploughing effect and material plastic side flow, along with micro tears caused by the built-up edge [35]. When the feed rate was minimal, the ploughing effect and BUE formation were severe in the cutting process. This caused uneven scratch marks and many micro tears on the machined surface morphology, resulting in a large surface roughness. However, since the feed rate increased to larger than 0.2 mm, these physical factors were weakened, but the theoretical residual area became significant in terms of the surface morphology and was a dominant factor for the surface roughness. The large theoretical residual area induced a large surface roughness. The minimum surface roughness was obtained with a feed rate of 0.15 mm/r.

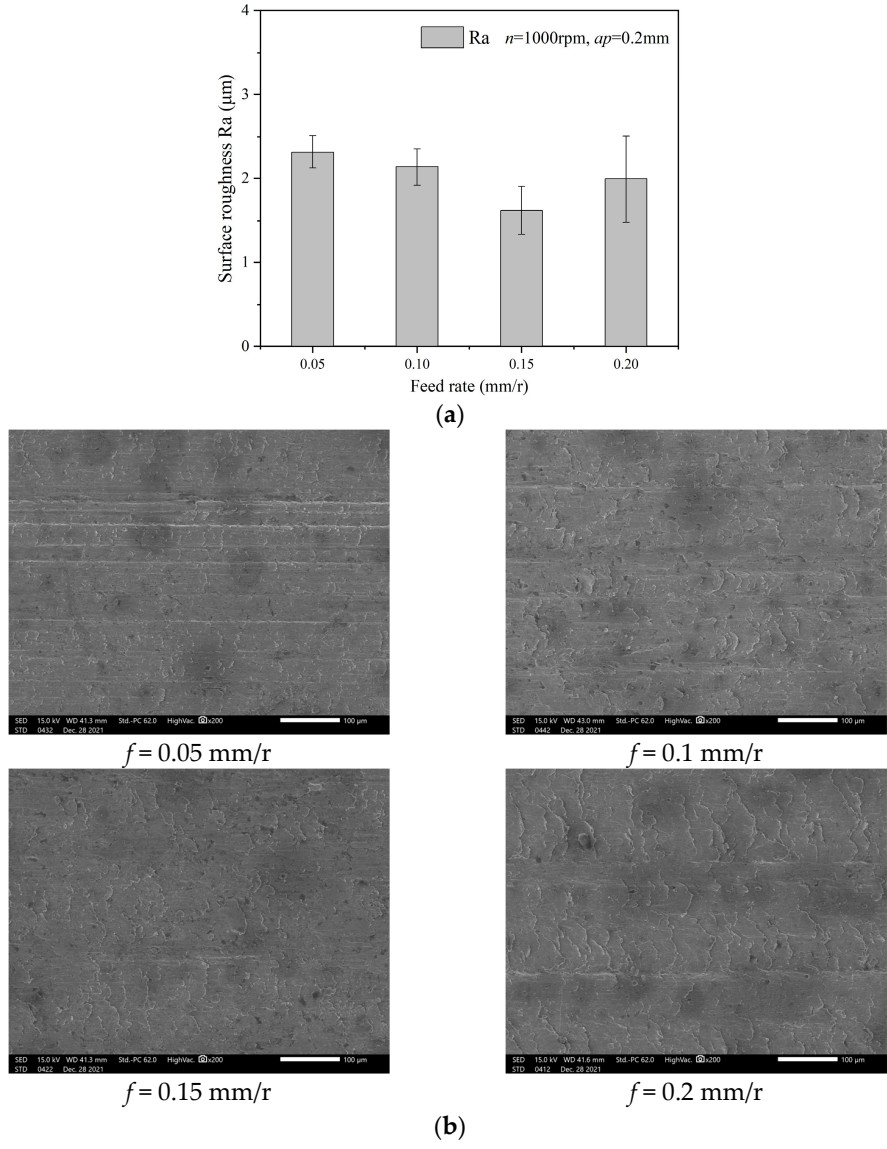

**Figure 7.** Effect of feed rate on surface quality: (**a**) effect of feed rate on surface roughness; (**b**) surface morphology varying with different feed rates.

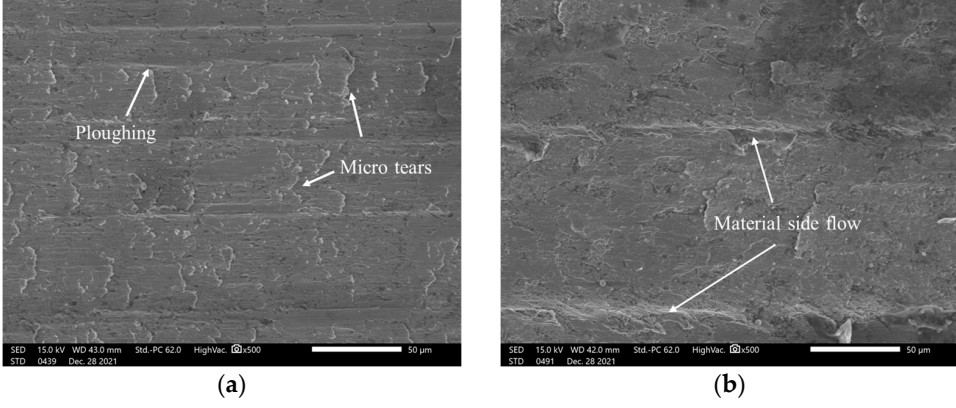

**Figure 8.** Enlarged view of surface morphology: (**a**) micro tears on surface morphology; (**b**) material side flow on surface morphology.

The variations of surface roughness and surface morphology with cutting speed are shown in Figure 9. It is shown that the surface roughness value gradually decreased with an increase in cutting speed. According to the surface morphology, the scratch marks on the machined surface decreased under the higher cutting speed, and a better surface roughness was obtained. This was probably due to the BUE formed in cutting process becoming the protrusion part in front of the tool tip. This led to the actual rake angle changing from a negative rake angle (equal to angle of tool chamfer) to a positive rake angle. This phenomenon improved the effective negative rake angle on the tool chamfer in the cutting process, which was equivalent to increasing the sharpness of the cutting edge. The sharper cutting edge could decrease the ploughing effect and material plastic side flow in the cutting process. Hence, more minor scratch marks were formed on the surface, as shown in Figure 8b.

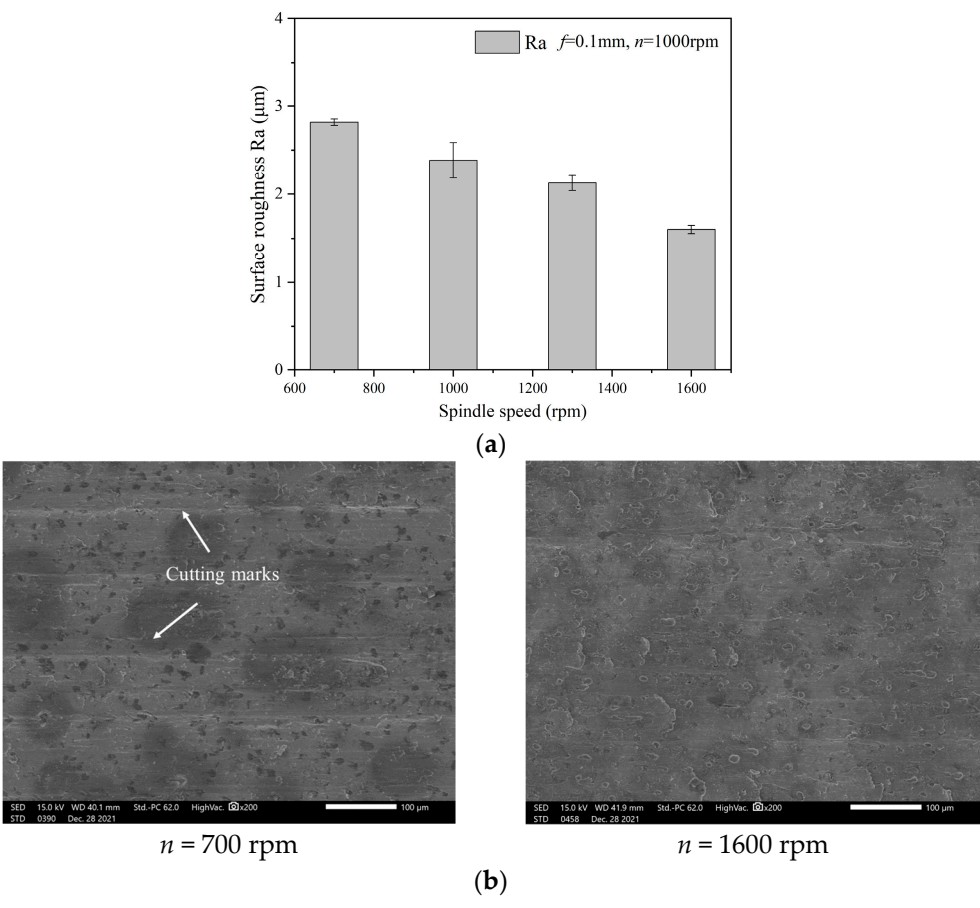

**Figure 9.** Effect of cutting speed on surface quality: (**a**) effect of cutting speed on surface roughness; (**b**) surface morphology varies with different cutting speed.

Figure 10 shows the surface roughness and surface morphology under different depths of cut. The surface roughness became larger with an increase in the depth of cut. The increasing rate was slow at first, before increasing when the depth of cut exceeded 0.3 mm. When the depth of cut was very small, the equivalent tool cutting edge angle was close to zero. The tool nose was helpful to squeeze the machined surface to be very flat, and a relatively low surface roughness was achieved. When the depth of cut reached 0.4 mm, on the basis of the surface morphology, it was found that the cutting marks became prominent, and scales appeared on the machined surface, which deteriorated the surface roughness.

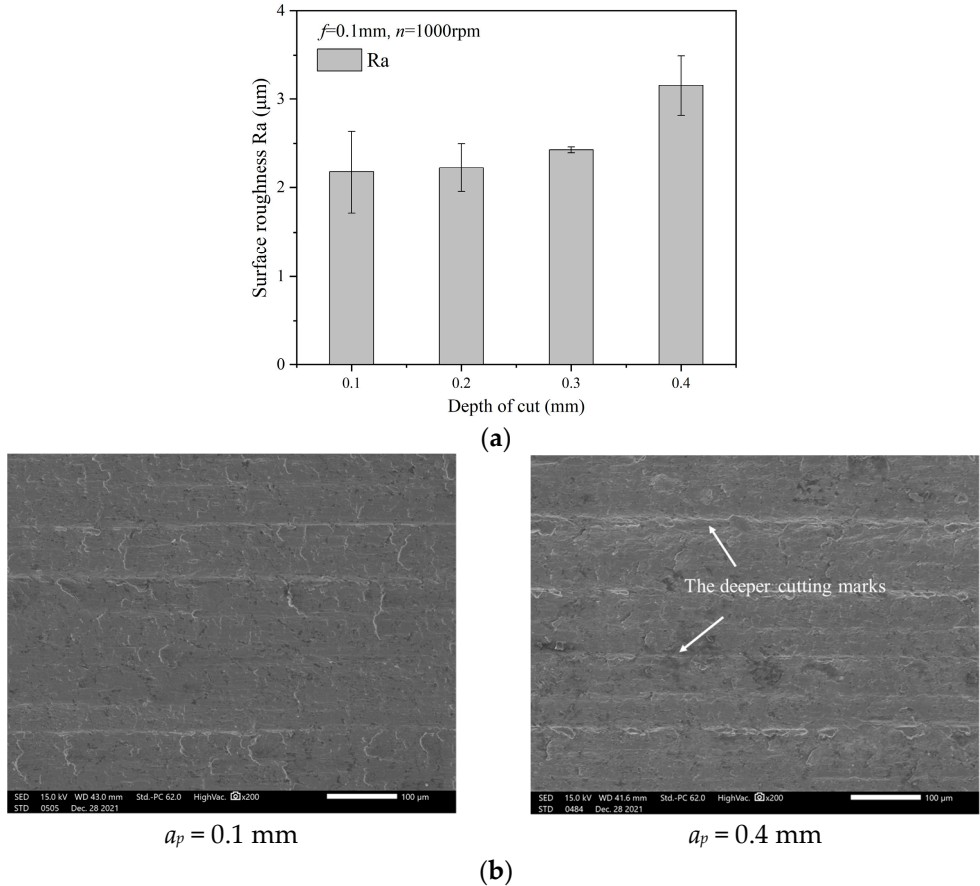

**Figure 10.** Effect of depth of cut on surface quality: (**a**) effect of depth of cut on surface roughness; (**b**) surface morphology varying with different depth of cuts.

### 3.4. The Tool Wear

Figure 11 shows the rake face and flank face morphology of the PCBN cutting tool with a cutting distance of 4 m. Compared with the original tool morphology in Figure 1, despite the high hardness difference between the tool material and gray cast iron, there was noticeable tool wear according to the tool morphology. The tool wear on the rake face was much more severe than on the flank face. It was found that the PCBN tool wear mode mainly involved the rake face wear, including some micro notches and many adhesion materials. The distribution of adhesion materials on the rake face was beyond the tool surface range that actually took part in the cutting process. Due to the severe adhesion material and micro chipping, the tool chamfer at the rake face was worn, rendering its observation challenging. However, there was minimal adhesion material on the flank face with hardly any tool wear. The observed tool wear characterizations are consistent with the literature [19].

To confirm the composition of the adhesion material, an EDS analysis was performed, and the results are shown in Figure 12. It was found that, in addition to the elements of the PCBN tool, the main elements of the adhesion material constituted Fe, Si, Mn, etc. These elements corresponded to the composition of gray cast iron, confirming that the adhesion material was the workpiece material. These adhesion materials on the tool surface could easily generate the built-up edge phenomenon in the cutting process.

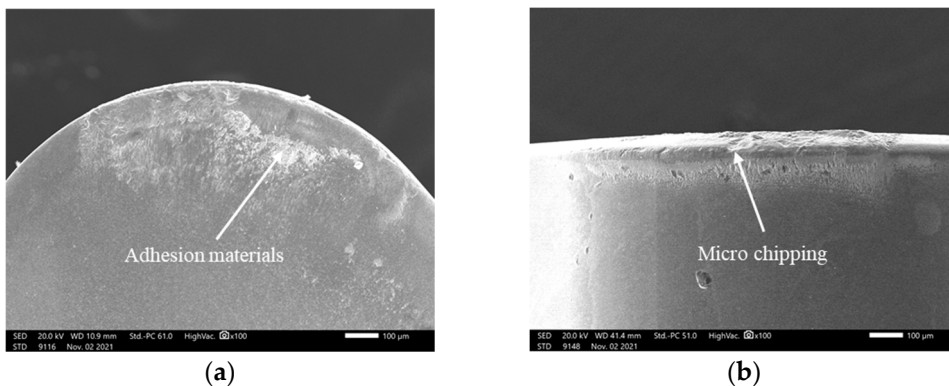

**Figure 11.** PCBN tool wear characteristics: (**a**) rake face; (**b**) flank face.

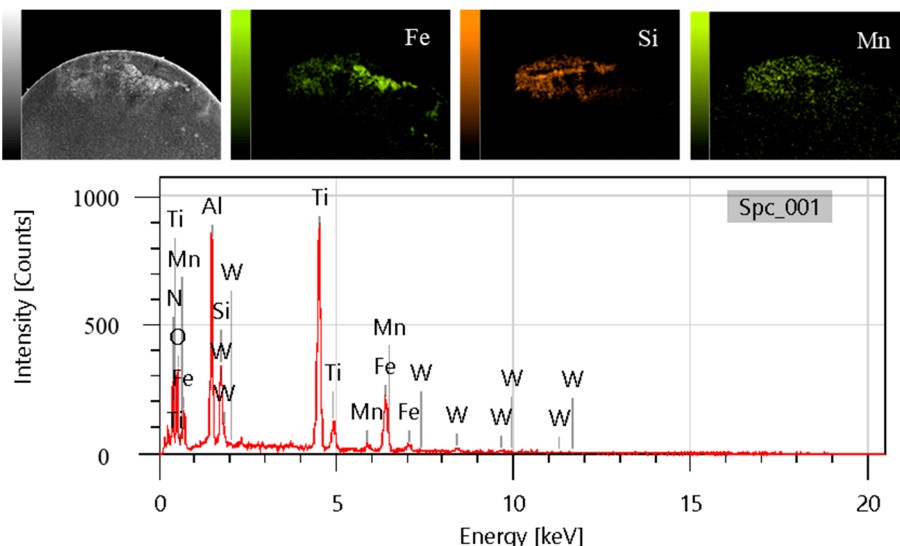

**Figure 12.** EDS analysis of adhesion material.

Figure 13 shows an enlarged view of the adhesion material on the rake face and flank face. It is revealed that the tool wear mechanism was mainly adhesion wear at the rake face and micro chipping at the tool chamfer. Under the action of the cutting force, the workpiece material and fragment chips were adhered and embedded on the tool surface, thus forming a bulge on the tool surface. These adhesion materials underwent serious friction and shear action with the continuous cutting process. Hence, some adhesion materials were fractured and fell off from tool surface. Simultaneously, this caused some tool material to spall and drop off from the surface, and then micro notches were formed on the rake face, as shown in Figure 13a. This adhesion wear mechanism would continue periodically throughout the cutting process. A negative rake angle caused very serious stress distribution in the local zone on the tool chamfer. At the same time, the adhesion wear could induce some micro cracks or notches on the too chamfer, which could further develop into micro chipping under a large stress distribution.

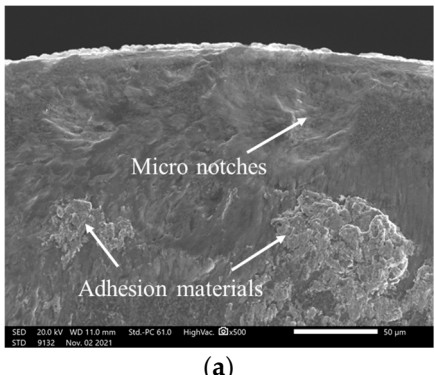 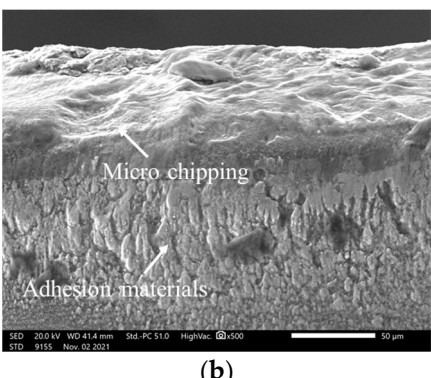

(**a**)  (**b**)

**Figure 13.** The adhesions on tool surface: (**a**) adhesion on rake face; (**b**) adhesion on clearance face.

## 4. Conclusions

This paper presented an experimental study on the drying turning of gray cast iron using a PCBN cutting tool. The following conclusions could be drawn:

1.  Both the cutting temperature and the cutting force increased with the increase in feed rate and depth of cut. Although the cutting temperature increased with an increase in cutting speed, the built-up edge during the cutting process induced the cutting force to gradually increase with a larger cutting speed.
2.  The surface roughness values were found to decrease at first, and then increase with an increase in feed rate; the minimum surface roughness value was obtained with a feed rate of 0.15 mm/r. The feed rate should be selected to exceed the tool chamfer width to obtain a good surface quality. The surface roughness gradually decreased with the increase in cutting speed, but increased with the increase in depth of cut. The equivalent tool cutting edge angle decreased at a low depth of cut, which was helpful to obtain a smooth surface and achieve a relatively low surface roughness.
3.  The PCBN tool wear mode mainly involved micro notches on the rake face and micro chipping on the tool chamfer. During the cutting process, the workpiece material could easily adhere to and embed onto the tool surface, while the fall off of adhesion materials induced some tool material spalling on the tool surface. Micro chipping and adhesion wear were the main wear mechanism for the PCBN tool due to the negative rake angle on the tool chamfer.

**Author Contributions:** Experiments and paper writing, G.Y. and X.W.; Measurement, J.S. and Z.W.; Data analysis, F.J. All authors have read and agreed to the published version of the manuscript.

**Funding:** This research was funded by the National Natural Science Foundation of China, grant number 52075097 (W.Z.).

**Institutional Review Board Statement:** Not applicable.

**Informed Consent Statement:** Not applicable.

**Data Availability Statement:** Not applicable.

**Conflicts of Interest:** The authors declare no conflict of interest.

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
