# Peer review of "Experimental Investigation on the Machinability of PCBN Chamfered Tool in Dry Turning of Gray Cast Iron"

_processes, doi:10.3390/pr10081547_

Round 1

Reviewer 1 Report

The topic of the article is interesting and I also see the possibility of transferring knowledge directly into industrial practice. The title is adequate to the contribution. The theoretical background is understandable. The content of the main section is presented clearly. The methodology is sufficient. In the experiment suitable measurement technology was used which contributes to the relevance of the data. The processing of results and conclusions is comprehensible and meets the standards of a scientific article. The supporting figures are appropriate. I consider that the article is suitable for publishing in the MDPI Processes, but some details are missing:

- Generally, the authors should present and highlight the scientific novelty. What is new about the results described in the article? Why was the research carried out and what is the result of it?

(Materials and Methods)

- Fig 2 - No markings for dynamometer, amplifier and computer program name.

- No detailed description of the cutting tools materials (symbols, chemical composition).

(Results and Discussion)

- The article could present information on a statistical analysis of the research results. For example, are there average values in the graphs (Figs. 4,6-10)? What was the scatter of the results or the standard deviation, etc. Is it possible to present a mathematical formula to determine the optimal values of the cutting parameters?

- The BUE phenomenon should be described in more detail. Is it possible to show the photos?

- In the Conclusions, point 1 should be supported by concrete details.

Author Response

-Generally, the authors should present and highlight the scientific novelty. What is new about the results described in the article? Why was the research carried out and what is the result of it? (Materials and Methods) Answer: The purpose of this work is to assess the machinability of PCBN chamfered tool in hard machining of gray cast iron. The effect of machining parameters on the cutting temperature, cutting force and surface quality was measured. The results are helpful foe the machining parameter selection in machining of brittle materials with PCBN cutting tools. - Fig 2 - No markings for dynamometer, amplifier and computer program name. Answer: The name and type of dynamometer and amplifier have been added in Fig. 2. - No detailed description of the cutting tools materials (symbols, chemical composition). Answer: The used cutting tool in this work was the PCBN cutting tool (grade BN500, Sumitomo). The gran size of CBN materials is about 1 ‒ 2 μm, and the volume proportion of CBN material in the PCBN tools is about 56% after measured. - The article could present information on a statistical analysis of the research results. For example, are there average values in the graphs (Figs. 4,6-10)? What was the scatter of the results or the standard deviation, etc. Is it possible to present a mathematical formula to determine the optimal values of the cutting parameters? Answer: The error bar has been added in the paper. - The BUE phenomenon should be described in more detail. Is it possible to show the photos? Answer: The BUE phenomenon mainly is reflected from the surface morphology defects and the adhesion materials on the tool surface. - In the Conclusions, point 1 should be supported by concrete details. Answer: The point 1 has been revised.

Reviewer 2 Report

This paper investigated the performance of PCBN chamfered tool in the hard machining of gray cast iron. The effects of machining parameters on the cutting temperature and cutting force are explored. The machined surface roughness and the tool wear are characterized. However, the discussions on machining mechanisms are not enough. This paper should be improved before being accepted. Please refer to the following comments:

11. The experimental results of this paper are clear, but lack of discussions. It is more like an experiment report than an article. The effect of chamfered tool on cutting temperature, cutting force, and surface quality need to be discussed more. Since the highlight of this paper is the use of chamfered tool, what is the difference between conventional tool and chamfered tool should be pointed out.

22. Line 192, “The probable reason is that when the feed rate is smaller than 0.1 mm/r, the actual undeformed chip thickness is less than tool chamfer width of 0.1 mm…” This explanation is not robust, because the actual undeformed chip thickness is related to not only feed rate, but also the depth of cut and tool nose radius. The effect of chamfer width is not well discussed.

I3. In conclusion part, line 304, authors concluded that “The feed rate should be selected to exceed the tool chamfer width to obtain good surface quality”. What happens if we keep the feed rate unchanged, and reduce the chamfer width? In this way, it seems that conventional tool can get a better surface quality when the feed rate is unchanged.

44. There some typing mistakes. For example, “temperture” in the title of sub-section 3.1.

Author Response

The experimental results of this paper are clear, but lack of discussions. It is more like an experiment report than an article. The effect of chamfered tool on cutting temperature, cutting force, and surface quality need to be discussed more. Since the highlight of this paper is the use of chamfered tool, what is the difference between conventional tool and chamfered tool should be pointed out.

Answer: Compared to the conventional tool, the tool chamfer is helpful to enhance the cutting edge, and avoid the premature chipping of tool cutting edge during machining of the brittle materials.

Line 192, “The probable reason is that when the feed rate is smaller than 0.1 mm/r, the actual undeformed chip thickness is less than tool chamfer width of 0.1 mm…” This explanation is not robust, because the actual undeformed chip thickness is related to not only feed rate, but also the depth of cut and tool nose radius. The effect of chamfer width is not well discussed.

Answer: You are right. The actual undeformed chip thickness is related to the feed rate, depth of cut and tool nose radius. We have revised this explanation.

In conclusion part, line 304, authors concluded that “The feed rate should be selected to exceed the tool chamfer width to obtain good surface quality”. What happens if we keep the feed rate unchanged, and reduce the chamfer width? In this way, it seems that conventional tool can get a better surface quality when the feed rate is unchanged.

Answer: From the surface roughness results, the minimum surface roughness is obtained when the feed rate exceeds the tool chamfer. In theory, the conventional tool without tool chamfer can obtain the sharper cutting edge, and then achieve the better surface roughness. However, this sharper cutting edge may be occurs the premature chipping in machining of brittle materials, and then results in the worse surface quality. In our opinion, there is an optimal tool chamfer width, and this is our future research direction. 

There some typing mistakes. For example, “temperture” in the title of sub-section 3.1.

Answer: The spelling mistake has been revised.

Round 2

Reviewer 2 Report

Some typing mistakes exist. Please check text carefully. 

For example, Page 2 Line 89.

Author Response

Some typing mistakes exist. Please check text carefully. 

Answer:  We have revised the whole manuscript carefully and tried to avoid any grammar or spelling error. We also have asked several colleagues who are skilled authors of English papers to check the language.